# Simulation of Wrinkling during Bending of Composite Reinforcement Laminates

**DOI:** 10.3390/ma13102374

**Published:** 2020-05-21

**Authors:** Jin Huang, Philippe Boisse, Nahiène Hamila, Yingdan Zhu

**Affiliations:** 1INSA-Lyon, LaMCoS CNRS, Université de Lyon, F-69621 Lyon, France; jin.huang@insa-lyon.fr (J.H.); nahiene.hamila@insa-lyon.fr (N.H.); 2Zhejiang Provincial Key Laboratory of Robotics and Intelligent Manufacturing Equipment Technology, Ningbo Institute of Material Technology and Engineering, Chinese Academy of Sciences, Ningbo 315201, China; y.zhu@nimte.ac.cn

**Keywords:** fabrics/textiles, multilayer, bending, wrinkling

## Abstract

When a thick laminate is subjected to bending, under certain boundary conditions, wrinkles may appear and develop due to the inextensibility of the fibers. Wrinkling is one of the most critical defects in composite manufacturing. Numerical simulation of the onset and growth of such wrinkles is an important tool for defining optimal process parameters. Herein, several bending experiments of thick laminates are presented. They were found to lead to severe wrinkling and delamination of different kinds. It is shown that the history of loading changed the developed wrinkles. Stress resultant shell finite elements specific to textile reinforcement forming show their relevance to provide, for these wrinkles induced by bending, results in good agreement with the experiments, both with regard to the onset of the wrinkles and to their development. This numerical approach was used to improve the understanding of the phenomena involved in wrinkling and to define the conditions required to avoid it in a given process.

## 1. Introduction

Composite material has become essential for applications where a high stiffness-to-weight ratio is a key design parameter. On the other hand, the fiber matrix composition of these materials renders their manufacturing processes complex. Process simulation tools that are gradually being developed need to make it possible to avoid the trial-and-error design of these processes. Software have been invented for composite forming [1,2,3,4,5,6,7,8,9,10,11] and resin flow [12,13,14,15,16,17,18,19]. In addition, simultaneous ply-forming of a multilayered composite has been found to increase manufacturing efficiency [20,21,22,23,24,25,26].

However, when a laminate is formed, its bending can lead to the development of wrinkles. These wrinkles are one of the most severe defects that can affect a composite and their presence generally makes the composite unsuitable. It is essential to determine, especially by simulation, the conditions under which these wrinkles do not appear. Multilayer forming can lead to many types of defects [27]. Specific studies on multilayer unidirectional (UD) prepreg consolidated under autoclave pressure have been carried out [28,29,30,31,32]. In the case of L-shape forming, the mechanisms of wrinkle formation and the influence of process parameters have been studied [25,33,34].

Experimental investigations of wrinkle-type defects generated by the forming of multilayers are numerous and the phenomena are well analyzed. On the other hand, numerical simulations of these aspects are scarce. The simulation of wrinkling due to the consolidation of thick composite parts has been presented by Belnoue et al. [35]. Each ply was modeled by a 3D finite element layer in contact with its neighbors with consideration of viscoelastic behavior [36]. Simplified models of this phenomenon have been proposed. [37,38]. The numerical analysis of creasing and folding of laminated paperboard has been carried out for a continuous 2D solid medium [39,40].

Although the causes of wrinkling during manufacture can be plentiful, fiber inextensibility associated with the bending of thick laminate can lead to excess fiber length and is a major cause of flaw [27]. This paper proposes a simulation of the wrinkling caused by the bending of thick laminates. Stacks of dry plies (without resin) have been considered. The objective was to analyze only the wrinkles caused by the excess length of the inextensible fibers due to bending of a thick stack. This phenomenon is often of first order in multilayer forming processes. It is an issue that concerns other scientific fields, in particular folding of rock structures [41,42,43] and creasing and folding of paperboard [40,44,45,46,47].

In the present work, several bending experiments corresponding to large rotations of the laminate ends led to significant wrinkling. Different boundary conditions and loading steps resulted several different wrinkle. The objective was to show that stress-resultant shell elements specific to textile reinforcements provide relevant simulations of the development of wrinkles and the associated delamination in the case of bending of thick stacks with a large number of layers. The presented experiments and simulations were repeatable and represented the phenomena related to the excess length of the plies during the bending of thick laminates.

## 2. Presentation of Experimental Analyses

Ten-layer stacks of carbon fabric G1151 (Hexcel) (Figure 1a,b) and 100 layers of paper (Figure 1c) were considered. From a straight position, these stacks were bent by a relative rotation of the ends from 0° to 90°.

Rotations and displacements of the stack ends were imposed during loading. Figure 1 shows that significant wrinkles developed in the studied cases, especially when the boundary conditions of the stack prevented relative slippage. This wrinkling caused delamination of the different layers. Section 4 analyzes the wrinkles in the case of symmetrical bending with clamped ends (Figure 1a). Section 5 examines the influence of pressure at the ends upon wrinkling and upon the development of a bevel (Figure 1b). Section 6 analyzes the effect of global buckling before bending upon the development of wrinkles (Figure 1c). When it comes to symmetrical bending, the orientation of the textile layers was studied in Section 4. It was shown that a ±45° orientation made it possible to avoid wrinkling.

The objective of these different experimental tests was both to highlight the wrinkling due to the bending of the stacks and to serve as validations of the stress resultant approach developed for the simulation of textile composite forming. The different tests that are analyzed in this article are of the same type. They all impose a relative rotation of 90° of the ends. Nevertheless, the displacements of these ends and the history of loading differed and the obtained deformations and wrinkles varied widely.

## 3. Stress Resultant Shell Approach

In a shell approach, the internal forces on a representative unit cell (RUC) of the woven fabric are assumed to be represented by the tensions T^11^ and T^22^, in the warp and weft directions, the in-plane shear moment C_s_ and the bending moments M^11^ and M^22^ (Figure 2). These stress resultants are the conjugates of axial elongations $\varepsilon_{11},\varepsilon_{22}$, in-plane shear γ, and curvatures $\chi_{11},\chi_{22}$. The internal virtual work on a unit cell can be written:
(1)$$\begin{array}{l} \delta W_{\mathrm{int}}^{RUC}=\delta W_{\mathrm{int}}^{tension}+\delta W_{\mathrm{int}}^{shear}+\delta W_{\mathrm{int}}^{bending}\\ \delta W_{\mathrm{int}}^{RUC}=\delta \varepsilon_{11}T^{11}L_{1}+\delta \varepsilon_{22}T^{22}L_{2}+\delta \gamma \;C_{s}+\delta \chi_{11}M^{11}L_{1}+\delta \chi_{22}M^{22}L_{2} \end{array}$$
where $L_{1}$ and $L_{2}$ are the lengths of the RUC in the warp and weft directions. The virtual work theorem for any virtual displacement field equal to zero on the boundary with prescribed displacement is expressed as:(2)$$\delta W_{ext}^{}-\sum_{NRUC}\delta W_{\mathrm{int}}^{RUC}=\delta W_{acc}^{}$$

Here, NRUC is the number of unit cells of the woven fabric under consideration and $\delta W_{ext}^{}$ and $\delta W_{acc}^{}$ are respectively the external virtual work and the virtual work of the acceleration quantities.

The mechanical behavior of the textile reinforcement is given by the relations between the stress resultant T^11^, T^22^, C_s_, M^11^, M^22^ and the strains $\varepsilon_{11},\;\varepsilon_{22}$, γ, $\chi_{11},\;\chi_{22}$. The mechanical behaviors can be coupled [48,49,50,51,52], but for simplicity and due to lack of experimental data, it was assumed that the behaviors were decoupled. Finite elements composed of woven unit cells have been implemented from Equation (1) [5,53,54]. This stress resultant approach is well suited for the simulation of textile reinforcement forming. In particular, it takes into account the bending behavior independently of the membrane behavior which is essential for fibrous reinforcements whose bending stiffnesses are not directly related to that of the membrane, as is the case for classical shells.

Other approaches for decoupling the bending and membrane stiffnesses of woven fabrics have been proposed [55,56,57,58,59]. Membrane approaches have been developed for the simulation of fabric draping, however, it has been shown that the simulation of wrinkles requires a good description of the bending behavior [60,61]. The use of standard shell finite elements is not suitable because the bending stiffness is deduced from that of the membrane, which gives rise to bending stiffness values much higher than those of textile reinforcements and disturbs the wrinkle analysis. Another advantage of the stress resultant approach (Equation (1)) is the consideration of the resultant efforts on the elementary cell which leads to the direct use of the specific tests that have been developed to determine the mechanical behavior of textile reinforcements. The picture-frame and bias extension tests give the shear moment C_s_ as a function of the shear angle γ [62,63,64,65], and the Peirce- and Kawabata-type bending tests measure the bending moment M^αα^ as a function of the curvature $\chi_{\alpha \alpha }$ [61,66,67,68]. Finally, the biaxial tensile tests give the stresses as a function of the two axial deformations [69,70,71,72].

In this modeling approach, the transverse compaction of the layers is not considered. This would require the use of 3D finite elements or solid-shell elements [73,74,75]. In the present study, these stress resultant shells were used efficiently to simulate composite laminate wrinkling during bending and the associated ply separation (up to 100 layers).

## 4. Symmetrical Bending with Clamped Ends

In this test, a laminated composite was subjected to bending due to a symmetrical 45° rotation of its two ends. Here, both ends of the stack were clamped. The displacements of the nodes in the two ends (50 mm) (shown in Figure 3a) are imposed. It is known that in this case the bending of a laminate leads to wrinkles. This was confirmed experimentally in the following experiments. If at least one of the ends was unclamped, the bending of the laminate would take place without wrinkling, because of possible slippage between the layers, which would give rise to a bevel (Figure 4).

In Figure 5, the stack consisted of 10 layers of G1151 textile reinforcement manufactured by Hexcel. This reinforcement was an interlock fabric shown in Figure 3b. It has been studied and used as a test reinforcement in several investigations concerning the draping of preforms [22,76,77,78,79] and in particular the ITOOL project [80]. It consisted of an interlock weaving of 6K carbon yarns and the thickness of each ply of the stack was 1.3 mm.

The mechanical properties of G1151 (tension, in-plane shear and bending) have been studied in different studies [68,76,77] and the properties used in the stress resultant shell approach are given in Table 1.

Simulations were carried out using Plasfib in-house explicit finite element software [5]. The three node shell elements used are stress resultant shell elements as introduced in Section 3. Their formulation is described in [5]. Applications in composite forming are presented in [60,77]. They are rotation free elements (without degrees of freedom of rotation) and use the position of neighbouring elements to determine the curvature [81,82]. The characteristics of finite element models are given in Table 2.

Figure 5 shows the symmetrical bending of a stack of ten layers of G1151 oriented at 0°–90°. Due to the thickness of the stack and the inextensibility of the fibers, a significant wrinkling developed with ply separation. The simulation of the symmetrical bending test in Figure 5a was performed and is shown in Figure 5b. Ten layers of resultant stressed shell elements were placed in contact and clamped at both ends. Figure 5b,c illustrate that the deformed shape obtained by the simulation was in good agreement with the experimental test and that it gave a good description of the wrinkles caused by the bending of the laminate.

The test analyzed in Figure 6 was the same as in Figure 5 with the exception that the stack was composed of 100 layers of paper. The deformation was of the same nature. The mechanical characteristics of the sheets of paper are given in Table 3 [57] and Figure 6b shows that the simulation satisfactorily predicted the wrinkles for all 100 layers. Table 4 compares the experimental and simulated values of the positions of the apex point A of the wrinkles.

In Figure 7, the symmetrical bending was the same as before (rotation of 45° at each end), but the stacking was made with 10 layers of G1151 oriented at ±45°. The result of this was very different since the deformed shape had no wrinkles. The longitudinal inextensibility constraint of the 0°–90° case no longer existed and the stack thus deformed without wrinkling. A shear angle of 8° was measured (Figure 7a) on the outer surface and −5° on the lower surface. The ability for shear deformation during bending of plies oriented at 45° was noted in the case of 3D interlock fabrics [83]. The simulation of the symmetrical bending of the stack of G1151 plies oriented at ±45° did not show any wrinkling (Figure 7b) and confirmed the shear angles in the upper and lower layers.

In the test presented in Figure 8, symmetrical bending was applied to a stack of 10 layers of G1151 oriented alternately at 0°–90° and ±45°. Figure 8a pointed at the development of wrinkles of the same magnitude as for a 0°–90° stack. However, the deformation was different since the 0°–90° plies had the same shape, but the ±45° plies also formed wrinkles due to their being influenced by the neighbouring layers. The result was a deformation where the layers were in pairs: a ply at 0°–90° dragged its neighbouring ±45° ply with it. This deformation consisting of sets of two layers was correctly obtained by the simulation.

The experimental and simulated values of the positions of the apex point A of the wrinkles (Figure 5, Figure 6 and Figure 8) in the different symmetrical bending cases presented in Section 4 are compared in Table 4.

Experimental and numerical values are comparable although there are some differences. However, it should be noted that even though the bending and wrinkling experiments are repeatable, they are nevertheless subject to a certain amount of dispersion, which may explain some of the differences. Table 5 presents the experimental values of the height h_AA′_ of the wrinkle at apex point A in the case of the experiment shown in Figure 5. (Symmetrical bending of a stack of ten plies of Hexcel G1151^®^). The three values of h_AA′_ given in this table have been measured for three different tests. The dispersion is less than 5% and is of the same order of magnitude as the differences between the experimental and numerical values. The dispersion in the other tests is of the same magnitude.

The choice of the number of elements (Table 2) in the model is based on a compromise between the precision and the good description of the shape of the wrinkles susceptible to develop on the one hand and the calculation time on the other hand. The simulations take into account the geometrical non-linearities and the frictional contacts are numerous, especially in the 100-layer laminate. The number of elements used and specified in Table 2 are minimal while still allowing a good description of the wrinkles. Figure 9 shows the result of a simulation of a symmetrical bending with a lower number of elements (240 elements in Figure 9a instead of 480 elements in Figure 9b). The mesh in Figure 9a is too coarse for this case and the wrinkles are not correctly described.

Figure 10 displays, in the case of symmetrical bending, the values of the bending moments and the curvatures in the lower ply for which the wrinkling is greatest. The maximum curvature is in the vicinity of the edge and is 0.04 mm^−1^, i.e., a radius of curvature of 25 mm.

## 5. L-Flange Forming

This section describes laminates of 10 layers of G1151 and 100 paper layers formed into an L-flange (Figure 11, Figure 12 and Figure 13) [84,85,86,87,88]. The displacements of the right end are imposed. The right end of the laminate was rotated 90° while the left end is kept horizontal, leading to an L-shape as shown in Figure 12 and Figure 13. The value of the prescribed displacement at this right end is shown in Figure 11. On the left end a load is imposed. The effect of the magnitude of the load is analysed in the present section.

The influence of the compaction force applied at the left end was studied. When this load was almost zero (Figure 12a and Figure 13a), the layers of the stack could slip relative to one another and a bevel developed at the free end of the laminate. This bevel was due to the difference in the inner and outer radius of the L-flange and the length of the layers, which remained constant. In this case, with zero compaction load, the L-flange forming could be carried out without wrinkling. Figure 13a shows that the bevel angle in the case of the stack of 100 layers of paper was close to 30°. When the compaction force at the left end increased (10 N, Figure 12b and Figure 13b), wrinkles developed in the horizontal part of the laminate. The bevel became partial and only affected the lower part of the stack. There was no more slippage between the plies in the upper part of the stack, which caused wrinkles to form due to the constant length of the layers.

As can be seen in Figure 12c and Figure 13c, the compaction force at the left end was sufficient to clamp the stack at least partially. The force increases to, F = 20 N, there was a very small relative slippage of the plies, there was no longer any bevel and the wrinkling became significant.

These tests confirmed that the L-flange forming of a laminate led to significant wrinkling if both ends were blocked. In order for the L-flange forming to prevent wrinkling, it is necessary that one end of the laminate creates a bevel. The simulations of the L-flange forming shown in Figure 9 and Figure 10 accurately described all aspects of both wrinkle formation and the total or partial development of the bevel. A simulation of the forming using the stress resultant shells presented in Section 3 proved to be a suitable tool to determine whether or not the forming conditions would lead to wrinkling.

Table 6 compares the experimental and simulated values of the positions of the apex point B of the wrinkles (Figure 12 and Figure 13) after L-flange forming presented in Section 5.

## 6. Bending after Buckling

This section analyses the bending of a laminate after global buckling. In this test, a first step consisted of buckling the stack of 100 paper layers due to imposed horizontal displacements of the ends (Figure 14a,b). In a second step, the laminate was bent by the rotation of the ends. Two cases were considered: in Figure 14c,e, the bending was symmetrical with end rotations of 45° each. Wrinkles developed because the ends of the stack were clamped. Two wrinkles appeared symmetrically on the left and right sides of the laminate. It can be noted that although the bending was symmetrical (rotation of 45° at each end), the fact that the rotation took place after the global buckling led to a different type of wrinkling than the one obtained in Section 4 (Figure 6). In the second case (Figure 14d,f), only the left end was rotated 45°. A single wrinkle developed on the right side of the laminate. Figure 14e,f show the simulation results of these two cases. They were in good agreement with the experiments.

Table 7. compares the experimental and simulated values of the positions of the point C, D, E and F apex of the wrinkles (Figure 14c,e) after bending after buckling.

Table 8 compares the experimental and simulated values of the positions of the point G and H apex of the wrinkles (Figure 14d,f) after buckling and rotation of a single side presented in Section 6.

## 7. Remarks and Conclusions

The thick laminate bending experiments presented in this article showed significant wrinkling and ply separation due to the excess length of certain plies. Simulations of these cases based on stress resultant shell elements specific to textile reinforcements provided results in good agreement with the experiments for large ply numbers (100 plies). It was found to be a simulation tool that could analyse the influence of the parameters and select them for wrinkle-free manufacturing. Several remarks can be made:

The various tests presented were similar, but the wrinkles that developed differed in appearance. In particular, the symmetrical bending described in Section 4 and the bending after buckling presented in Section 6 both subjected the laminate to a 45° rotation of both ends. However, the wrinkle types were not the same. This was due to the non-linearities with regard to geometry and contact causing the pre-buckling (described in Section 6) to lead to two zones of wrinkling while only one developed in the case put forward in Section 4.

The experiments and simulations presented were quite consistent and characteristic of a given situation.

From the point of view of real production, the analyses presented show that the simulation of wrinkling during a folding operation is possible with numerical models of acceptable complexity for a number of plies that can be significant. From the point of view of the laminate quality, wrinkles are not acceptable. Simulations such as those presented above should be used to define the conditions of the manufacturing process to avoid these wrinkles.

If we consider the bending laminates as a single shell for the whole laminate (modelling with a single shell in the thickness), the position of the normals in the set of deformed configurations (plotted the different figures of the article) corresponded neither to Kirchhoff’s theory nor to Mindlin’s theory, thus confirming [57,61].

In this article, only the effect of excess fibre length induced by bending was considered. This is an important phenomenon that can lead to major defects. Nevertheless, there exist many potential sources of defects during the manufacturing of a composite of which the main ones are listed in [27]. For a given process it will be necessary to model all those that are likely to occur.

Only dry plies (without resin) were considered. This is the case for LCM processes where the resin is injected after the forming of a preform. However, a large part of composites are made by draping thermoset or thermoplastic prepregs and the resin plays an important role in possible defects of the process.

## Figures and Tables

**Figure 1 materials-13-02374-f001:**
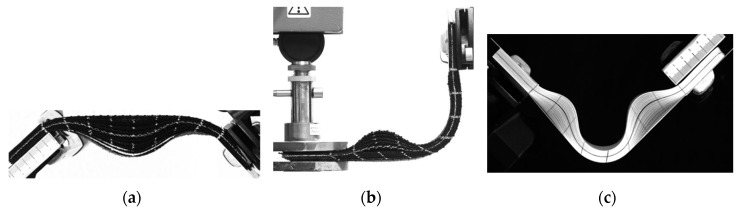
Experimental analysis of wrinkling during bending. (**a**) Symmetrical bending; (**b**) L-flange forming; (**c**) Bending after buckling.

**Figure 2 materials-13-02374-f002:**
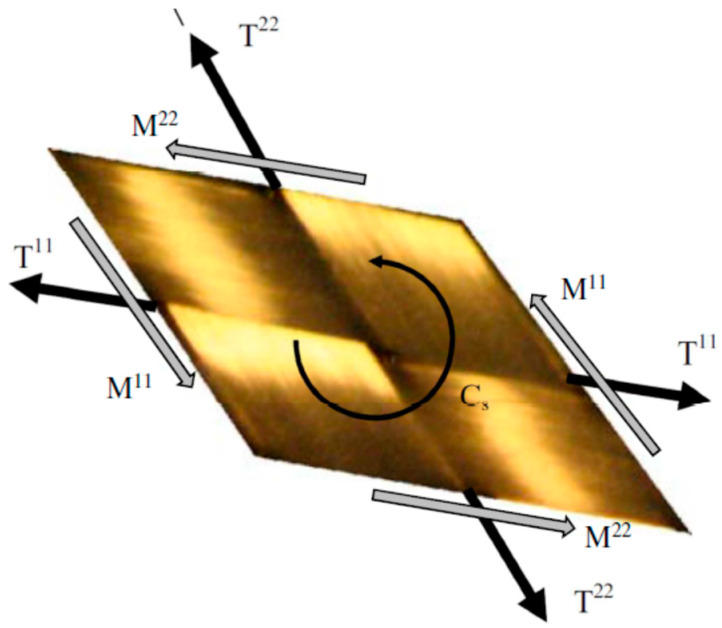
Stress resultants on a unit woven cell.

**Figure 3 materials-13-02374-f003:**
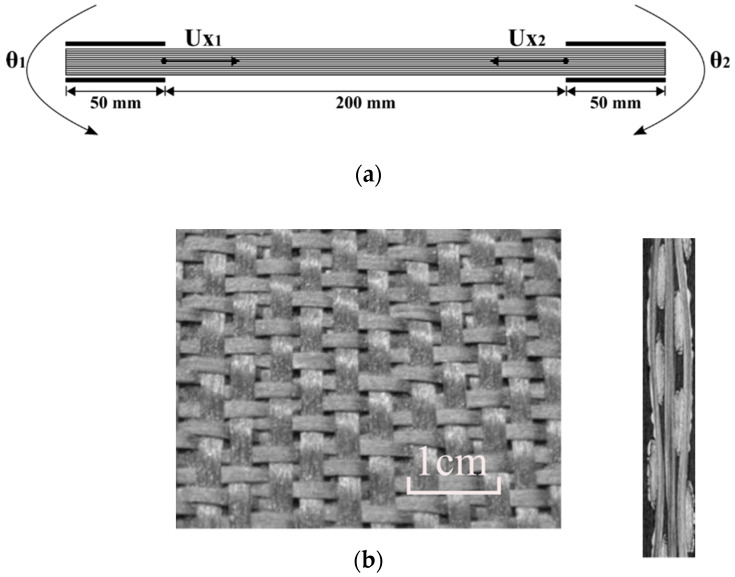
Symmetrical bending. (**a**) Initial state and prescribed rotations and displacements; (**b**) Hexcel G1151^®^.

**Figure 4 materials-13-02374-f004:**
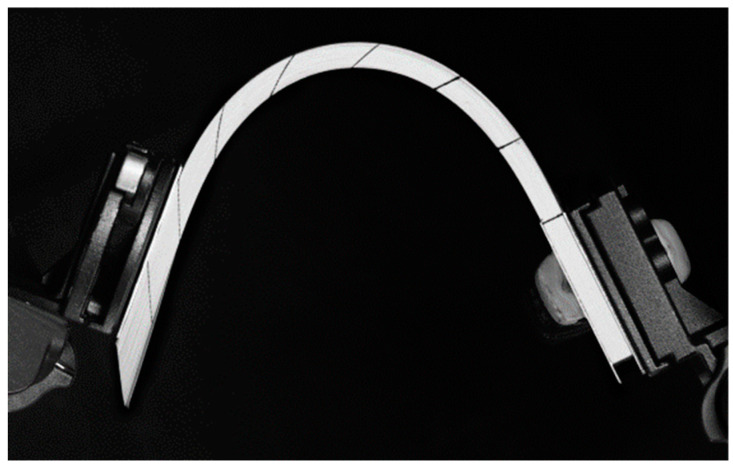
Bending with only one end clamped.

**Figure 5 materials-13-02374-f005:**
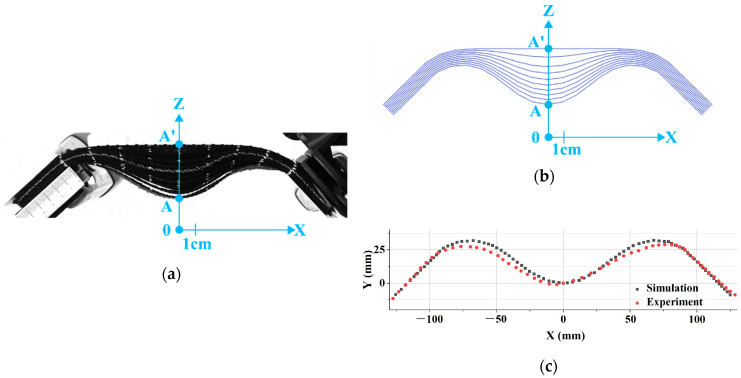
Symmetrical bending of a stack of ten plies of Hexcel G1151^®^. (**a**) Deformed shape obtained by experiment; (**b**) Deformed shape obtained by simulation; (**c**) Comparison of the experimental and numerical deflection for the inferior ply.

**Figure 6 materials-13-02374-f006:**
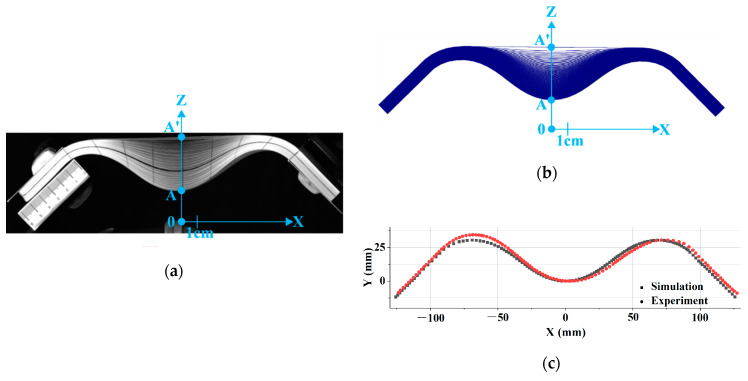
Symmetrical bending of a stack of 100 layers of paper. (**a**) Deformed shape obtained by experiment; (**b**) Deformed shape obtained by simulation; (**c**) Comparison of the experimental and numerical deflection for the inferior ply.

**Figure 7 materials-13-02374-f007:**
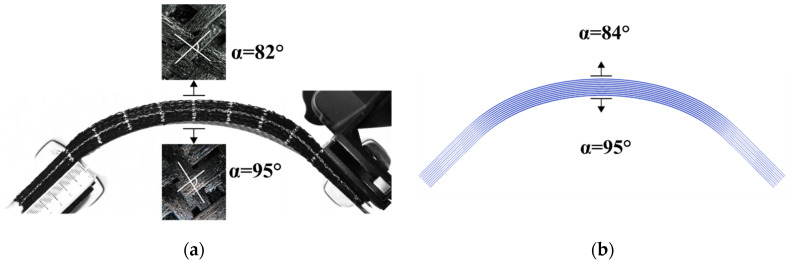
Symmetrical bending of a stack of ten plies of Hexcel G1151^®^ oriented at ±45° (**a**) Deformed shape obtained by experiment. (**b**) Deformed shape obtained by simulation.

**Figure 8 materials-13-02374-f008:**
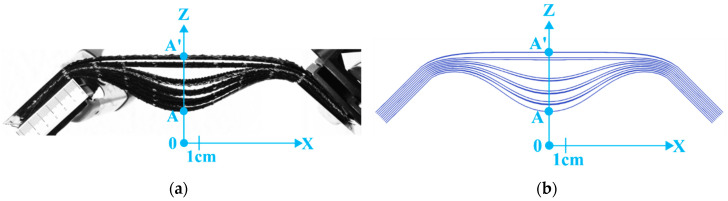
Symmetrical bending of a stack of ten plies of Hexcel G1151^®^ oriented successively at 0°–90° and 45°. (**a**) Deformed shape obtained by experiment; (**b**) Deformed shape obtained by simulation.

**Figure 9 materials-13-02374-f009:**
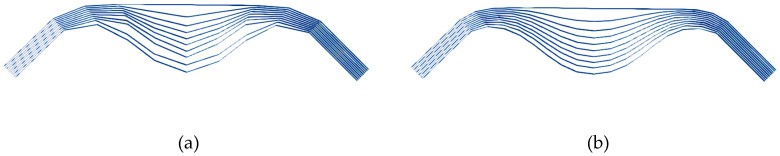
Symmetrical bending of a stack of ten plies of Hexcel G1151^®^. Comparison of the result of the deformed shape obtained when using (**a**) 240 and (**b**) 480 three node shell elements.

**Figure 10 materials-13-02374-f010:**
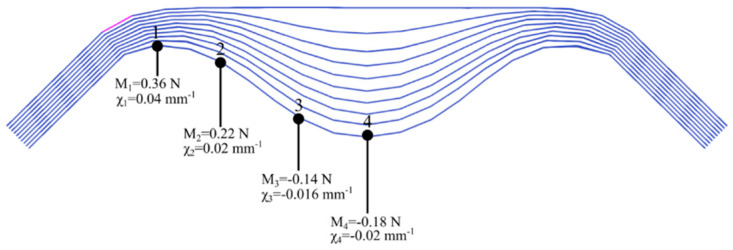
Symmetrical bending of a stack of ten plies of Hexcel G1151^®^. Bending moments and the curvatures in the lower ply.

**Figure 11 materials-13-02374-f011:**
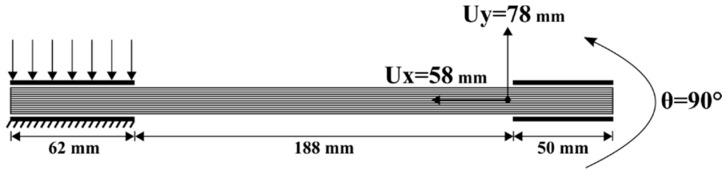
L-Flange forming. Initial state and prescribed rotation and displacements.

**Figure 12 materials-13-02374-f012:**
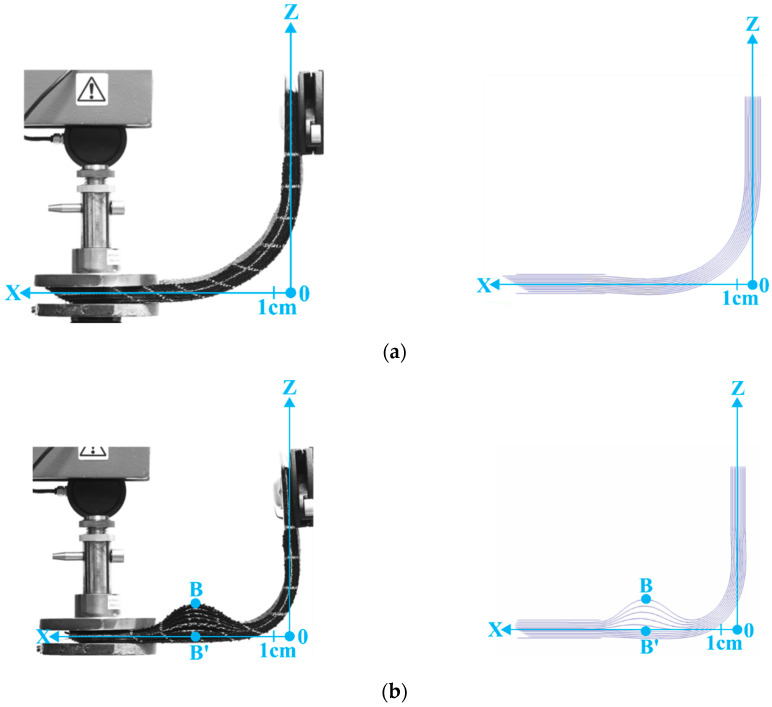
L-Flange forming. Initial state and prescribed rotation and displacements. L-Flange forming of a stack of ten plies of Hexcel G1151^®^. (**a**) L-Flange forming with a 1 N load on the left end. Experiment and simulation; (**b**) L-Flange forming with a 10 N load on the left end. Experiment and simulation; (**c**) L-Flange forming with a 20 N load on the left end. Experiment and simulation.

**Figure 13 materials-13-02374-f013:**
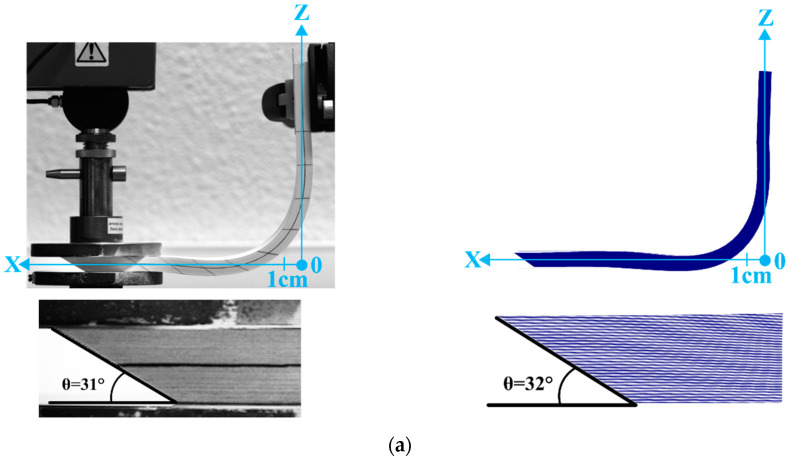
L-Flange forming of a stack of 100 layers of papers with comparison of the experimental and numeric bevel. (**a**) L-Flange forming with a 1N load on the left end. Experiment and simulation; (Zoom on the bevel at left end); (**b**) L-Flange forming with a 10 N load on the left end. Experiment and simulation; (**c**) L-Flange forming with a 20 N load on the left end. Experiment and simulation.

**Figure 14 materials-13-02374-f014:**
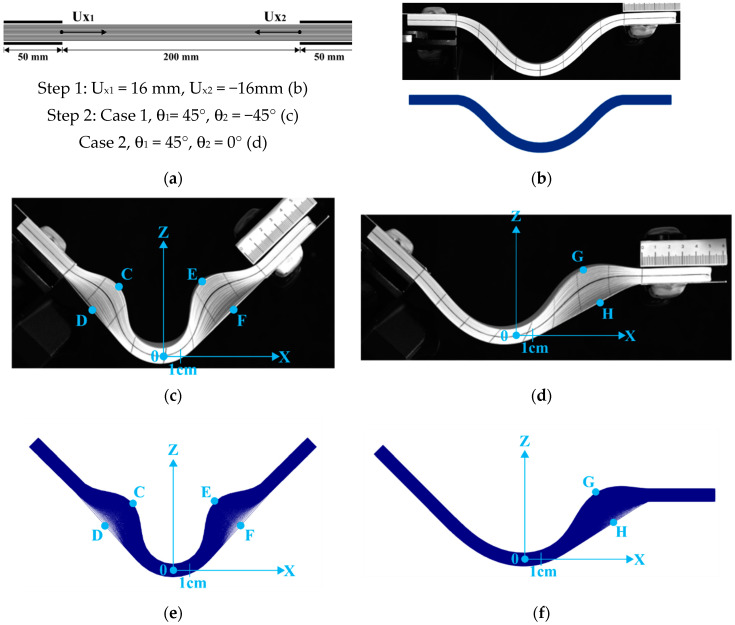
(**a**) Initial position; (**b**) First step: global buckling; (**c**) Second step symmetrical bending; (**d**) Second step: rotation of a single side; (**e**) Symmetrical bending: simulation; (**f**) Rotation of a single side: simulation.

**Table 1 materials-13-02374-t001:** Mechanical properties of the G1151^®^ fabric.

Tensile stiffness in warp and weft direction: $$T^{\alpha \alpha }=K_{\alpha }\varepsilon_{\alpha \alpha }$$	$K_{\alpha }=2300\;N/yarn\;(\alpha =1,2)$
In-plane shear stiffness: $$C_{s}\left(\gamma \right){=k}_{1}{\gamma +k}_{3}\gamma^{3}{+k}_{5}\gamma^{5}$$	$$k_{1}=0.37\;N\;\mathrm{mm}$$ $$k_{3}=-0.84\;N\;\mathrm{mm}$$ $$k_{5}=1.03\;N\;\mathrm{mm}$$
Bending stiffness in warp and weft direction: $$M^{\alpha \alpha }{=B}_{\alpha }\chi_{\alpha \alpha }$$	$B_{\alpha }=8.84\;N\;\mathrm{mm}\;(\alpha =1,2)$
Coefficient of friction between plies: Thickness of a ply	μ = 0.21 e = 1.3 mm

**Table 2 materials-13-02374-t002:** Characteristics of Finite Element Models.

Simulation	Number of Layers	Number of Elements per Layer	Number of Nodes per Layer	Number of Degree of Freedom per Layer	Total Number of Degree of Freedom
Stack of G1151 layers	10	48	50	150	1500
Stack of paper layers	100	96	98	294	29,400

**Table 3 materials-13-02374-t003:** Mechanical properties of a layer of paper.

**Young’s modulus E:**	4.61 GPa
**Shear modulus G**	1.92 GPa
**Coefficient of friction between plies:**	μ=0.20
**Thickness of a ply**	e = 0.1 mm

**Table 4 materials-13-02374-t004:** Comparison of the experimental and numerical height h_AA′_ of the wrinkle at apex A (mm).

	Figure 5	Figure 6	Figure 8
Experiment h_AA′_	44.6	42.8	46.6
Simulation h_AA′_	45.6	40.1	44.7

**Table 5 materials-13-02374-t005:** Measure of the height h_AA′_ of the wrinkle at apex point A for three different tests (mm).

	Test 1	Test 2	Test 3
Experiment h_AA′_	44.6	46.2	45.3

**Table 6 materials-13-02374-t006:** Comparison of the experimental and numerical position of the apex point B (X, Z) and height h_BB′_ of the wrinkle after L-Flange forming (mm).

	Figure 12b	Figure 12c	Figure 13b	Figure 13c
Experiment B	(78.5, 27.4)	(73.1, 31.6)	(93.4, 20.5)	(90.9, 31.9)
Simulation B	(78.2, 26.3)	(75.1, 30.1)	(88.9, 22.9)	(83.0, 31.0)
Experiment h_BB′_	27.4	31.6	20.5	31.9
Simulation h_BB′_	26.3	30.1	22.9	31.0

**Table 7 materials-13-02374-t007:** Comparison of the experimental and numerical position of the points C(X, Z), D(X, Z), E(X, Z), F(X, Z) and height of the wrinkle after bending after buckling (mm).

	Position C	Position D	Position E	Position F	h_CD_	h_EF_
Experiment	(−32.4, 50.8)	(−51.8, 33.7)	(28.1, 54.6)	(50.8, 33.8)	26.2	31.3
Simulation	(−30.2, 50.0)	(−51.0, 33.3)	(30.8, 52.0)	(50.4, 33.4)	27.0	26.4

**Table 8 materials-13-02374-t008:** Comparison of the experimental and numerical position of the point G(X, Z) and H(X, Z) and height of the wrinkle after buckling and rotation of a single side (mm).

	Position G	Position H	h_GH_
Experiment	(49.9, 48.6)	(62.3, 24.4)	28.5
Simulation	(52.9, 50.3)	(66.2, 27.6)	26.4

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
