# Peer review of "Simulation of Wrinkling during Bending of Composite Reinforcement Laminates"

_materials, 2020, doi:10.3390/ma13102374_

Round 1

Reviewer 1 Report

The paper refers to experiment and simulation of wrinkling during bending of laminated composites. The presented results ore interesting giving a tool for designing high quality laminates. I recommend the paper for publication but after some revision. The comments and suggestions are listed below:

1. Line 34: the shortcut “UD” should be explained before the first usage.

2. There is not any details about the software used and other calculation parameter.

3. In my opinion, all the simulation results should be presented in a spacial coordinates and scale.

4. Please, include details about exact geometry of the tested and simulated layers (thickness etc…).

5. Please, extend the discussion in relation to significance of the obtained results in real production processes as well as possible improvement of the laminate quality.

Author Response

The authors wish to thank the reviewers for the useful comments and suggestions that were essential to improve the quality of our work. Response to the reviewer’s comments is given below.

The English language has been corrected by a society of prof readings. A Proofreading certificate is joined to the cover letter.

In the revised manuscript, changes are highlighted with blue in the text

6 tables and 4 figures have been added in order to better quantify results and numerical-experimental comparisons.

Review 1

Comments and Suggestions for Authors

The paper refers to experiment and simulation of wrinkling during bending of laminated composites. The presented results ore interesting giving a tool for designing high quality laminates. I recommend the paper for publication but after some revision. The comments and suggestions are listed below:

Thank you for the comment.

  1. Line 34: the shortcut “UD” should be explained before the first usage.

Author’s response: Added in line 34 (introduction):

Specific studies on multilayer unidirectional (UD) prepreg consolidated under autoclave pressure have been carried out [28-32].

  1. There is not any details about the software used and other calculation parameter.

Author’s response: Added in section 4 (line 146):

Simulations were carried out using PLASFIB in-house explicit finite element software [5]. The three node shell elements used are stress resultant shell elements as introduced in section 3. Their formulation is described in [5]. Applications in composite forming are presented in [60, 77]. They are rotation free elements (without degrees of freedom of rotation) and use the position of neighbouring elements to determine the curvature [81, 82]. The characteristics of finite element models are given in Table 2.

Table 2. Characteristics of Finite Element Models (Added in section 4)

  1. In my opinion, all the simulation results should be presented in a spacial coordinates and scale.

Author’s response:

Axes and scales have been added on each figure.

Tables 4,5,6,7 and 8 have been added to compare the positions of experimental and simulated apex points.

  1. Please, include details about exact geometry of the tested and simulated layers (thickness etc…).

Author’s response:

Geometries have been completed and are given in Figure 3, 11, 14. Thickness, number of plies and finite element models are given in tables 1, 2, 3.

  1. Please, extend the discussion in relation to significance of the obtained results in real production processes as well as possible improvement of the laminate quality.

Author’s response: Added in the conclusion:

From the point of view of real production, the analyses presented show that the simulation of wrinkling during a folding operation is possible with numerical models of acceptable complexity for a number of plies that can be significant. From the point of view of the laminate quality, wrinkles are not acceptable. Simulations such as those presented above should be used to define the conditions of the manufacturing process to avoid these wrinkles.

Reviewer 2 Report

The introduction and background sections are overly vague. More details related to specific results of references and the relevance to the current article should be provided.

Lines 66-72 – The author reverences the sub pictures of figure 1 out of order AND references future sections out of order. It is confusing.

Line 75 – “The analyzed tests were close enough” what does that mean? Close enough to what? What is the measure for “close enough”.

Line 96 – “the mechanical behaviors can be coupled [..] it was assumed that the behaviors were decoupled” Which mechanical behaviors is the author referring to? Is everything decouple or is this in-plane and out-of-plane behavior, as is standard?

Line 106-109 – This is an incorrect assessment of the state of the art of simulation. Current, commercially available tools decouple membrane and bending behavior when dealing with fabrics. This is possible in Aniform, Abaqus, etc…

Line 118-119: The use of the term laminate and use of delamination is difficult in the context of dry sheets. Lamination implies initial bonding. These sheets are not. Perhaps ply separation?

Figure 3 is stated to be a description of their experiments. However, Figure 3C is actually an image of an alternate experiment which allows slippage of plies. Figure 3C should be separated as it’s own figure, and Figure 3 should be remade with a proper diagram of the experimental setup.

Line 125-127 –The author needs to specify that they are studying the formation of wrinkles under boundary conditions that are known to PRODUCE wrinkles. The description of the experiments has been imprecise until now.

Lines 141-142 – The author shows the results of a simulation, but they provide no details about the simulation. What were the boundary conditions (displacement controlled, load controlled, etc…)? You say the layers were clamped at both ends, were they layers clamped by strong normal force and friction interactions or some other boundary condition that behaves similar to the clamp? Was the solution methodology implemented in a commercial software or was a custom solve developed? If a custom software, what checks were performed for verification? Is it a 2 dimensional solution? Results as presented seem to indicate that each layer was simply a line. What was the mesh density? What was the pre-deformed geometry? I know nothing about the simulation besides a “stress resultant” shell element was used. How am I to believe the results when I know nothing about it?

Lines 150 – The author presents a new experiment but paper was the material system. A subset of material properties were provided for the paper, but not as many as for the composite material system earlier. What did you do for the friction coefficients? Did you change the ply thickness?

Section 5 – Simulation and experimental setup were not discussed well. The only description of the experiment is provided Figure 8 which can hardly count as an experimental discussion.

Figure 9 seems to indicate that the boundary conditions in the simulation did not match that of the experiments as there is significant slippage of plies (very apparent in 9C). This figure is later described to show that there was no slippage. When clearly the top and bottom plies are in 1 position and the bulk of the rest have moved.

Section 6 – It would be nice for the author to actually differentiate between the wrinkle types observed and comment on the conditions that produced each type. It would also be useful to have a quantitative comparison between the experiment and simulation results besides a cross sectional view and plot. Did any simulation perform better or worse than the others? Why is that?

Figure 11 should show the simulated bucking shape before end rotation.

Conclusions: This article essentially only considered visual similarity as a metric of simulation quality. This should be clarified in the abstract, introduction, and conclusions.

Author Response

The authors wish to thank the reviewers for the useful comments and suggestions that were essential to improve the quality of our work. Response to the reviewer’s comments is given below.

The English language has been corrected by a of prof reading society.

A Proofreading certificate is joined to the letter.

In the revised manuscript, changes are highlighted with blue in the text

In the revised manuscript, changes are highlighted with blue in the text

6 tables and 4 figures have been added in order to better quantify results and numerical-experimental comparisons.

  1. Review 2.

Comments and Suggestions for Authors

  1. The introduction and background sections are overly vague. More details related to specific results of references and the relevance to the current article should be provided.

Author’s response:

The objective of this paper is to present experiments of composite reinforcement laminates bending that lead to wrinkling and a finite element approach based on stress resultant shell elements specific to textile reinforcements that provides an efficient simulation of the onset and development of wrinkles.

As stated in the introduction, experimental investigations of wrinkle-type defects generated by the forming of multilayers are quite numerous and the phenomena are fairly well analyzed. But there are few numerical simulations of these phenomena. The simulation of wrinkling due to the consolidation of thick composite parts has been presented by Belnoue et al. Each ply was modeled by a 3D finite element layer in contact with its neighbors.

Added in the abstract:

Stress resultant shell finite elements specific to textile reinforcement forming show their relevance to provide, for these wrinkles induced by bending, results in good agreement with the experiments, both with regard to the onset of the wrinkles and to their development.

  1. Lines 66-72 – The author reverences the sub pictures of figure 1 out of order AND references future sections out of order. It is confusing.

Author’s response:

The order of presentation of the figures has been changed

Added in section 2:

Section 4 analyzes the wrinkles in the case of symmetrical bending with clamped ends (Figure 1a).

  1. Line 75 – “The analyzed tests were close enough” what does that mean? Close enough to what? What is the measure for “close enough”.

Modified in section 2 (line 76):

The different tests that are analyzed in this article are of the same type. They all impose a relative rotation of 90° of the ends. Nevertheless, the displacements of these ends differed and the obtained deformations and wrinkles varied widely.

  1. Line 96 – “the mechanical behaviors can be coupled [..] it was assumed that the behaviors were decoupled” Which mechanical behaviors is the author referring to? Is everything decouple or is this in-plane and out-of-plane behavior, as is standard?

Author’s response: The considered mechanical behaviors are bending, shear and tensile behaviors.

Some couplings between these behaviors of a textile reinforcement have been highlighted by several studies.

For example:

In plane-shear–tension coupling: Ref. 48, 50, 51

In-plane shear-bending coupling: Ref 49

In-plane shear-bending-tension-friction coupling: Ref. 52

Nevertheless, for simplicity and due to lack of experimental data, and models, it was assumed that the behaviors were decoupled.

  1. Line 106-109 – This is an incorrect assessment of the state of the art of simulation. Current, commercially available tools decouple membrane and bending behavior when dealing with fabrics. This is possible in Aniform, Abaqus, etc…

Author’s response: In section 3 (line 108):

The use of standard shell finite elements proposed in commercial software is not suitable

has been replaced by:

The use of standard shell finite elements is not suitable

Author’s response:

  1. Line 118-119: The use of the term laminate and use of delamination is difficult in the context of dry sheets. Lamination implies initial bonding. These sheets are not. Perhaps ply separation?

Author’s response:

 “delamination” has been replaced throughout the article by “ply separation”.

  1. Figure 3 is stated to be a description of their experiments. However, Figure 3C is actually an image of an alternate experiment which allows slippage of plies. Figure 3C should be separated as it’s own figure, and Figure 3 should be remade with a proper diagram of the experimental setup.

Author’s response:

Figure 3c was transformed into an independent figure 4.

  1. 125-127 –The author needs to specify that they are studying the formation of wrinkles under boundary conditions that are known to PRODUCE wrinkles. The description of the experiments has been imprecise until now.

Author’s reponse:

Added in section 4 (line 131):

In this test, a laminated composite was subjected to bending due to a symmetrical 45° rotation of its two ends. Here, both ends of the stack were clamped. It is known that in this case the bending of a laminate leads to wrinkles. This was confirmed experimentally in the following experiments.

  1. Lines 141-142 – The author shows the results of a simulation, but they provide no details about the simulation. What were the boundary conditions (displacement controlled, load controlled, etc…)? You say the layers were clamped at both ends, were they layers clamped by strong normal force and friction interactions or some other boundary condition that behaves similar to the clamp? Was the solution methodology implemented in a commercial software or was a custom solve developed? If a custom software, what checks were performed for verification? Is it a 2 dimensional solution? Results as presented seem to indicate that each layer was simply a line. What was the mesh density? What was the pre-deformed geometry? I know nothing about the simulation besides a “stress resultant” shell element was used. How am I to believe the results when I know nothing about it?

Author’s response:

Line 131. In section 4, (Symmetrical Bending with Clamped Ends). The displacements of the nodes in the two ends (50mm) (shown in figure 3) are imposed.

 Line 237. In section 5 (L-flange Forming): The displacements of the right end are imposed. The right end of the laminate was rotated 90° while the left end is kept horizontal, leading to an L-shape as shown in figures 12 and 13. The value of the prescribed displacement at this right end is shown in Figure 11. On the left end a load is imposed. The effect of the magnitude of the load is analyzed in the present section.

Line 271. In section 6, the displacements of the nodes in the two ends are imposed. In this test, a first step consisted of buckling the stack of 100 paper layers due to imposed horizontal displacements of the ends.

These precisions have been added to the different sections.

Information on the finite element models and software have been added line 146 to 153 and in Table 2,

There is no pre-deformed geometry.

The deformed laminates are drawn in the software Paraview and the shell elements appear in the form of a line. Nevertheless they have a thickness that is taken into account both in mechanical calculation and in contact management.

The shell elements used are 3D. Most of the cases analysed in this study can be represented in one plane, except for the case of Figure 7, (bending of a stack oriented at ±45°)

where in-plane shear in the plies plays a major role and is therefore clearly 3-D.

  1. Lines 150 – The author presents a new experiment but paper was the material system. A subset of material properties were provided for the paper, but not as many as for the composite material system earlier. What did you do for the friction coefficients? Did you change the ply thickness?

Author’s response:

The paper is considered as isotropic. Its properties are given by the Young modulus and Poisson ratio. The thickness of a paper ply was 0.1 mm (the thickness of a G1151 ply is 1.3 mm).These thicknesses are given in table 1 and 3.

     Coulomb friction coefficient are 0.21 (G1151) and 0.2 (paper). They are given in table 1 and 3.

  1. Section 5 – Simulation and experimental setup were not discussed well. The only description of the experiment is provided Figure 8 which can hardly count as an experimental discussion.

Author’s response:

Line 236. Added in section 5: This section describes laminates of 10 layers of G1151 and 100 paper layers formed into an L-flange (Figures 11 to 13) [84-88]. The displacements of the right end are imposed. The right end of the laminate was rotated 90° while the left end is kept horizontal, leading to an L-shape as shown in figures 12 and 13. The value of the prescribed displacement at this right end is shown in Figure 11. On the left end a load is imposed. The effect of the magnitude of the load is analyzed in the present section.

  1. Figure 9 seems to indicate that the boundary conditions in the simulation did not match that of the experiments as there is significant slippage of plies (very apparent in 9C). This figure is later described to show that there was no slippage. When clearly the top and bottom plies are in 1 position and the bulk of the rest have moved.

Author’s response:

The slippage between the plies is important in fig. 12a (F=1N) and a bevel appears (both in the experiment and in the simulation).

This slip decreases as the force on the left end increases . There is only a very small slippage in 12c (F= 20N). This slippage is very small and of the same nature in both the experiment and the simulation.

Line 255. Added in section 5:

As can be seen in Figure 12c and 13c, the compaction force at the left end was sufficient to clamp the stack at least partially. The force increases to, F=20N, there was a very small relative slippage of the plies, there was no longer any bevel and the wrinkling became significant.

  1. Section 6 – It would be nice for the author to actually differentiate between the wrinkle types observed and comment on the conditions that produced each type. It would also be useful to have a quantitative comparison between the experiment and simulation results besides a cross sectional view and plot. Did any simulation perform better or worse than the others? Why is that?

Author’s response:

The wrinkles that develop in the different cases analyzed in this study are all of the type ‘excess fiber length’ as defined in [27]. The fibers are quasi inextensible and the bending of the laminate constrains certain fibers to buckle.

Five tables (4,5,6,7,8) have been added that compare the position of the experimental and numerical apex of the wrinkles.

Line 202. Added in section 4:

Experimental and numerical values are comparable although there are some differences. However, it should be noted that even though the bending and wrinkling experiments are repeatable, they are nevertheless subject to a certain amount of dispersion, which may explain some of the differences.

Table 5 (have been added) presents the experimental values of the height hAA’ of the wrinkle at apex point A in the case of the experiment shown in Figure 5. (Symmetrical bending of a stack of 10 plies of Hexcel G1151®). The three values of hAA’ given in this table have been measured for three different tests. The dispersion is less than 5% and is of the same order of magnitude as the differences between the experimental and numerical values. The dispersion in the other tests is of the same magnitude.

  1. Figure 11 should show the simulated bucking shape before end rotation.

Author’s response:

A figure showing the bucking shape before end rotations has been added in Figure 14b.

  1. Conclusions: This article essentially only considered visual similarity as a metric of simulation quality. This should be clarified in the abstract, introduction, and conclusions.

Author’s response:

Fives tables (4,5,6,7,8) have been added that compare the position of the experimental and numerical apex of the wrinkles. Axes and scales have been added in each figure.

Reviewer 3 Report

1 The title “ Simulation of wrinkling during bending of laminated composites” is suggested to be “Simulation of wrinkling during bending of fabric laminates ”. Because it is only fiber fabric without resin.

2 In numerical simulation, it is suggested that some necessary information should be provided like element and node number, element and node number, convergence and element l size independence.

3 How do you consider the friction role among the fibers or fabrics?

4 In numerical simulation about bending with buckling, it is suggested that the maximum stress or maximum strain should be given.

5 If possible, please give some descriptions about the critical condition about wrinkles and buckling in bending test and simulation.

Author Response

The authors wish to thank the reviewers for the useful comments and suggestions that were essential to improve the quality of our work. Response to the reviewer’s comments is given below.

The English language has been corrected by a proof reading society.

A  Proofreading certificate is joined to the letter.

In the revised manuscript, changes are highlighted with blue in the text

6 tables and 4 figures have been added in order to better quantify results and numerical-experimental comparisons.

  1. Review 3

Comments and Suggestions for Authors

  1. The title “ Simulation of wrinkling during bending of laminated composites” is suggested to be “Simulation of wrinkling during bending of fabric laminates ”. Because it is only fiber fabric without resin.

Author’s response:

The tittle has been modified to

“Simulation of wrinkling during bending of composite reinforcement laminates

  1. In numerical simulation, it is suggested that some necessary information should be provided like element and node number, element and node number, convergence and element l size independence.

Author’s response:

The table 2. ‘Characteristics of Finite Element Models’, has been added that gives the number of nodes and elements in different simulations

Concerning the fineness of the mesh, the following text has been added in section 4 and the Figure 9 has been added.

Line 212, added to section 4:

The choice of the number of elements (Table 2) in the model is based on a compromise between the precision and the good description of the shape of the wrinkles susceptible to develop on the one hand and the calculation time on the other hand. The simulations take into account the geometric non-linearities and the frictional contacts are numerous, especially in the 100-layer laminate. The number of elements used and specified in table 2 are minimal while still allowing a good description of the wrinkles. Figure 9a shows the result of a simulation of a symmetrical bending with a lower number of elements (240 elements in Figure 9a instead of 480 elements in Figure 9b). The mesh in Figure 9a is too coarse for this case and the wrinkles are not correctly described.

Added Figure 9. Symmetrical bending of a stack of 10 plies of Hexcel G1151®. Comparaison of the result of the deformed shape obtained when using 240 and 480 three node shell elements

How do you consider the friction role among the fibers or fabrics?

The presented simulations are macroscopic. Each ply is modelled by a layer of shell elements. The contact between the plies is controlled by a penalty method associated with a Coulomb friction model. Coefficient values for the G1151 fabric and for the paper layers are given in Tables 1 and 3 (0.21 and 0.2). They are taken from previous studies on G1151 and from the literature for paper.

  1. In numerical simulation about bending with buckling, it is suggested that the maximum stress or maximum strain should be given.

Author’s response:

Line 223. Added in section 4:

Figure 10 displays, in the case of symmetrical bending, the values of the bending moments and the curvatures in the lower ply for which the wrinkling is greatest. The maximum curvature is in the vicinity of the edge and is 0.04 mm-1 , i.e. a radius of curvature of 25 mm.

Added : Figure 10. Symmetrical bending of a stack of 10 plies of Hexcel G1151®. Bending moments and the curvatures in the lower ply

  1. If possible, please give some descriptions about the critical condition about wrinkles and buckling in bending test and simulation.

Author’s response:

In the experiments, the wrinkles are deterministic and repeatable given the boundary conditions. As shown in Table 5, there is nevertheless some dispersion in the deformed geometry (in the order of 5%). For simulations, the approach is explicit and the solution is that of a dynamic problem without instability and critical condition.

The wrinkles that develop in the different cases analyzed in this study are all of the type ‘excess fiber length’ as defined in [27]. The fibers are quasi inextensible and the bending of the laminate constrains certain fibers to buckle.

Round 2

Reviewer 2 Report

The authors have satisfactorily addressed my comments. Their effort in responding to the quick turn around required by the journal is appreciated.